# Time of day is associated with paradoxical reductions in global signal fluctuation and functional connectivity

**Csaba Orban**[1,2,3,4]*, **Ru Kong**[1,2,3], **Jingwei Li**[1,2,3], **Michael W. L. Chee**[2,3,5,7], **B. T. Thomas Yeo**[1,2,3,5,6,7]*

**1** Department of Electrical and Computer Engineering, N.1 Institute for Health and Memory Networks Program, National University of Singapore, Singapore, **2** Centre for Sleep and Cognition, Yong Loo Lin School of Medicine, National University of Singapore, Singapore, **3** Clinical Imaging and Research Centre, Yong Loo Lin School of Medicine, National University of Singapore, Singapore, **4** Neuropsychopharmacology Unit, Centre for Psychiatry, Imperial College London, London, United Kingdom, **5** Centre for Cognitive Neuroscience, Duke-NUS Medical School, Singapore, **6** Martinos Center for Biomedical Imaging, Massachusetts General Hospital, Charlestown, Massachusetts, United States of America, **7** NUS Graduate School for Integrative Sciences and Engineering, National University of Singapore, Singapore

* eleco@nus.edu.sg (CO); thomas.yeo@nus.edu.sg (BTTY)

**Data Availability Statement:** Data are from the Human Connectome Project S1200, which is publicly available at https://www.humanconnectome.org/. Data points underlying

## Abstract

The brain exhibits substantial diurnal variation in physiology and function, but neuroscience studies rarely report or consider the effects of time of day. Here, we examined variation in resting-state functional MRI (fMRI) in around 900 individuals scanned between 8 AM and 10 PM on two different days. Multiple studies across animals and humans have demonstrated that the brain's global signal (GS) amplitude (henceforth referred to as "fluctuation") increases with decreased arousal. Thus, in accord with known circadian variation in arousal, we hypothesised that GS fluctuation would be lowest in the morning, increase in the midafternoon, and dip in the early evening. Instead, we observed a cumulative decrease in GS fluctuation as the day progressed. Although respiratory variation also decreased with time of day, control analyses suggested that this did not account for the reduction in GS fluctuation. Finally, time of day was associated with marked decreases in resting-state functional connectivity across the whole brain. The magnitude of decrease was significantly stronger than associations between functional connectivity and behaviour (e.g., fluid intelligence). These findings reveal time of day effects on global brain activity that are not easily explained by expected arousal state or physiological artefacts. We conclude by discussing potential mechanisms for the observed diurnal variation in resting brain activity and the importance of accounting for time of day in future studies.

## Introduction

Circadian rhythms govern diverse aspects of physiology, including sleep/wake cycles [1], cognition [2], gene expression [3], temperature regulation [4], and endocrine signalling [5]. Similarly, studies of brain function in both humans and animals have documented time of day–

Figs 1 and 3 and S1, S3, S4, S8, S9, and S10 Figs are presented in S1 Data, and data points underlying Fig 2 and S2 Fig are presented in S2 Data. The list of participants who passed our visual quality screening of their pulse and respiratory data can be found at the GitHub repository maintained by the Computational Brain Imaging Group at https://github.com/ThomasYeoLab/CBIG/tree/master/stable_projects/preprocessing/Orban2020_tod.

**Funding:** We are currently supported by Singapore Ministry of Education Tier 2 (MOE2014-T2-2-016), National University of Singapore (NUS) Strategic Research (DPRT/944/09/14), NUS School of Medicine Aspiration Fund (R185000271720), Singapore National Medical Research Council (NMRC/STaR/015/2013; CBRG/0088/2015), NUS Young Investigator Award, and the Singapore National Research Foundation (NRF) Fellowship (Class of 2017). Our research also utilized resources provided by the Center for Functional Neuroimaging Technologies, NIH P41EB015896 and instruments supported by NIH 1S10RR023401, NIH 1S10RR019307, and NIH 1S10RR023043 from the Athinoula A. Martinos Center for Biomedical Imaging at the Massachusetts General Hospital. Our computational work was partially performed on resources of the National Supercomputing Centre, Singapore (https://www.nscc.sg). Data were provided by the Human Connectome Project, WU-Minn Consortium (Principal Investigators: David Van Essen and Kamil Ugurbil; 1U54MH091657) funded by the 16 NIH Institutes and Centers that support the NIH Blueprint for Neuroscience Research; and by the McDonnell Center for Systems Neuroscience at Washington University. The funders had no role in study design, data collection and analysis, decision to publish, or preparation of the manuscript.

**Competing interests:** The authors have declared that no competing interests exist.

**Abbreviations:** BOLD, blood oxygen level–dependent; CBF, cerebral blood flow; DAN, dorsal attention network; DMN, default mode network; DVARS, temporal derivative of root mean square variance over voxels; EEG, electroencephalography; FD SD, standard deviation of framewise displacement; FDR, false discovery rate; fMRI, functional MRI; GS, global signal; GSR, global signal regression; HCP, Human Connectome Project; HR RMSSD, root mean of the successive differences of heart rate; HR SD, standard deviation of heart rate; LFP, local field potential; Lim, limbic network; MEG, magnetoencephalography; OLS, ordinary least squares; $pCO_2$, end-tidal $CO_2$ partial

dependent variation at multiple scales of brain organization. Examples include diurnal variation in synaptic protein levels and synaptic strength [6–8], the amplitude and slope of cortical or motor-evoked responses [6,9,10], and theta power [11,12], as well as changes in cerebral blood flow (CBF) [13–15], task-related blood oxygen level–dependent (BOLD) activation [16–18], resting-state BOLD signal amplitude [19–21], and resting-state functional connectivity (RSFC) [14,21–23].

Despite the clear influence of circadian rhythms on physiology, most studies of brain function do not report or consider the impact of time of day on their findings. Perhaps, there is an implicit assumption in the field that diurnal variation of brain activity is relatively small and unlikely to introduce substantial systematic bias into group analyses. Furthermore, most previous studies of time of day effects have employed small samples, within-participant designs with few time points, a controlled prestudy routine, and in some cases, sleep deprivation (e.g., [9,17]). Previous resting-state functional MRI (fMRI) studies that have focused on time of day have yielded inconsistent results, potentially because of small sample sizes [14,19–23]. Here, we exploited resting-state fMRI data from the Human Connectome Project (HCP) [24] involving (1) a large sample of 942 individuals scanned using cutting-edge multiband acquisition sequences, (2) a wide spread of scan times with a relatively smooth distribution between 9 AM and 9 PM, (3) test-retest data with varying scan times between two different days (sessions) enabling within-participant analyses, and (4) individually curated respiratory and cardiac recordings for examination of nonneural effects.

The HCP dataset allowed us to investigate how the magnitude of the brain's global signal (GS) fluctuation (also referred to as amplitude in the literature) varies throughout the day. Our focus on the GS was motivated by converging evidence of a strong link between GS fluctuation and arousal/vigilance levels in both humans and nonhuman primates (reviewed in [25]), which is known to change throughout the day [26]. Specifically, the magnitude of GS fluctuation (i.e., its temporal standard deviation) has been found to vary as a function of behavioural and electrophysiological indices of drowsiness [27–29] to increase following sleep deprivation [30] and to decrease following caffeine consumption [31]. Therefore, we hypothesised that levels of GS fluctuation would be lowest in the late morning hours (e.g., 11 AM), when arousal levels are normally at their peak [26,32–34]. In addition, we predicted that GS fluctuation would be elevated during midafternoon hours (e.g., 3 PM corresponding to the "mid-afternoon dip") and reduced in the early evening hours (e.g., 8–9 PM corresponding to the "wake maintenance zone"), when participants typically experience reduced and elevated states of arousal, respectively [26,32–34].

The contributions of this study are multifold. First, we demonstrated counterintuitive reductions in the magnitude of GS fluctuation from late morning to late evening across participants in two separate sessions. Our observation of high levels of GS fluctuation (typically a feature of low arousal) in the late morning and the gradual reduction in GS fluctuation throughout the day was inconsistent with expected patterns of circadian variation in arousal levels. Second, these findings were replicated within participants, across two sessions, thus ruling out individual differences in chronotype as a potential explanation. Third, the effect of time of day on GS fluctuation was not accounted for by differences in head motion, heart rate, and respiratory variability. Fourth, the reduction in GS fluctuation was accompanied by a widespread decrease in interregional functional connectivity larger than typical association strength between behaviour (e.g., fluid intelligence) and functional connectivity. In summary, this study revealed robust, diurnal changes in large-scale resting brain activity that merit further investigation.

pressure; ROI, region of interest; RSFC, resting-state functional connectivity; RV mean, mean of respiratory variation; RV SD, standard deviation of respiratory variation; Sal, salience network; TP, temporal parietal network.

## Results

### GS fluctuation declines with time of day

We considered ICA-FIX denoised resting-state fMRI data with MSMAll registration from the HCP S1200 release [24,35]. Although there are different definitions of GS in the literature, they all produce highly similar estimates [35–37]. Here, we defined the GS to be the mean time series obtained by averaging the preprocessed fMRI time courses across all cortical grey matter locations [37,38]. We also verified that our main findings were robust to using other definitions of GS, e.g., averaging signal across the whole brain. Based on earlier work, we defined GS fluctuation as the temporal standard deviation of the GS [31,27]. We adopt the term GS fluctuation instead of "GS amplitude" that is used in some previous studies.

Across participants, GS fluctuation was negatively correlated with time of day for session 1 (r = −0.17, p < 1.0e-05, N = 942; Fig 1A) and for session 2 (r = −0.20, p < 1.0e-05, N = 869; Fig 1B). Although individuals are known to express variation in circadian phase and chronotype [39], on average, participants should exhibit higher levels of arousal during late morning (e.g., at 11 AM) and again in the early evening (e.g., 8–9 PM) than in the midafternoon (e.g., at 3 PM), based on previous studies [26,32–34]. Given the literature suggesting a strong relationship between GS fluctuation and arousal levels, reviewed in [25], one would expect GS fluctuation to be lower in the late morning (when arousal is generally higher) than in the midafternoon (when arousal is generally lower). Instead, we found GS fluctuation to exhibit a steady decrease between participants from 11 AM to 9 PM (Fig 1A and 1B).

To test for within-participant effects, GS fluctuation and time of day from session 2 were subtracted from their respective values from session 1, and the resulting values were correlated. Within-participant differences in GS fluctuation as a function of within-participant differences in time of day (r = −0.20, p < 1.0e-05, N = 831) strongly corroborated the between-participant effects (Fig 1C).

Participants exhibited substantial between-participant variation in GS fluctuation at each time of day (see S1 Fig for a visualisation of scatterpoint density). Nevertheless, hourly (black dots in Fig 1A and 1B) or 3-hourly (black dots in Fig 1C) time-window averages of GS fluctuation closely followed the line of best fit, suggesting a linear decline that was not preferentially driven by any specific time of day. From 9 AM to 9 PM, the magnitude of GS fluctuation decreased from 0.22% of mean BOLD signal fluctuation to 0.17% (on average) in both sessions 1 and 2.

### Scanning run and time of day have opposing effects on GS fluctuation

Within a given session (day), participants were scanned for two consecutive runs. As shown in Fig 1D, GS fluctuation was significantly elevated in run 2 compared with that in run 1 within both session 1 (t = 13.0, p < 1.0e-05, N = 942) and session 2 (t = 9.0, p < 1.0e-05, N = 869), consistent with previous, similar analyses of the HCP dataset [41].

Nevertheless, the negative correlation between time of day and GS fluctuation was significant for each run in both sessions (S2 Fig). Thus, in summary, we observed two opposing effects: a fast increase of GS fluctuation occurring on the scale of minutes (i.e., run effect), superimposed on a more gradual decrease of GS fluctuation occurring on the scale of hours (i.e., time of day effect).

### Head motion and cardiac measures show strong correlation with GS fluctuation but not with time of day

GS fluctuation has been shown to correlate with run-level summary metrics of head motion, cardiac, and respiratory variables across participants [42]. To explore whether these factors

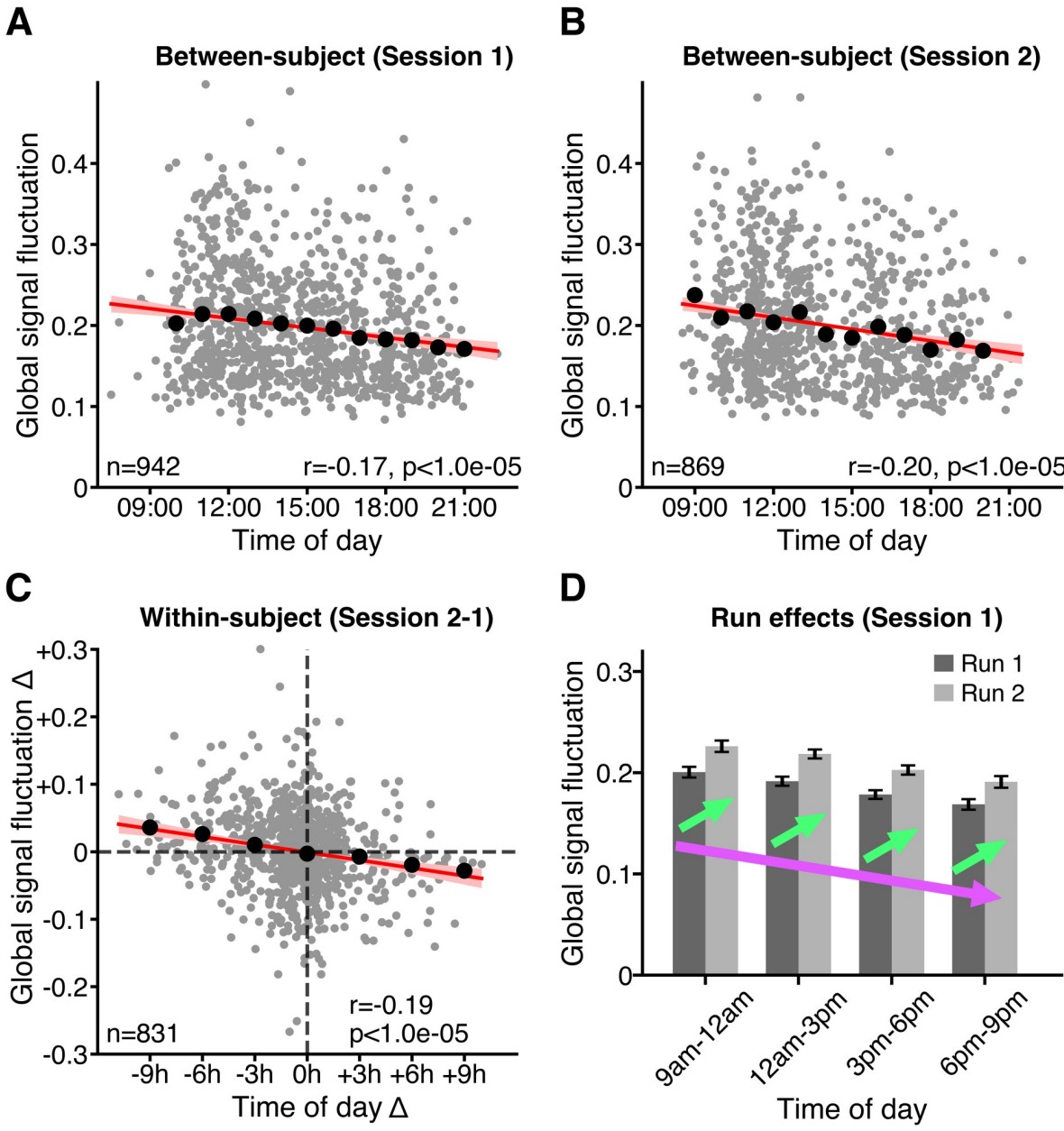

**Fig 1. The brain's GS fluctuation (standard deviation of GS) decreases with time of day. (A)** Between-participant variation in GS fluctuation as a function of time of day in session 1. **(B)** Between-participant variation in GS fluctuation as a function of time of day in session 2. **(C)** Within-participant variation in GS fluctuation Δ as a function of time of day Δ, where Δ denotes difference between session 2 and session 1. Grey dots denote individual participants. Black dots show mean of GS fluctuation in (A, B) hourly or (C) 3-hourly time windows. Line of best fit (red) was calculated based on data from all participants in each plot. Confidence interval is shown in light red. R values denote Pearson r correlation coefficient. *p*-Values were derived from 100,000 permutations, while keeping family structure intact [40]. **(D)** GS fluctuation is elevated in run 2 compared with that in run 1 despite downward shift in GS fluctuation as a function of time of day. Bar plots denote mean GS fluctuation across participants within 3-hourly time windows for each run. Error bars denote standard error of the mean. Two opposing effects are observable: a fast increase in GS fluctuation on the scale of minutes—i.e., run effect (green arrows)—superimposed on a downward drift of GS fluctuation occurring on the scale of hours—i.e., time of day effect (violet arrow). See S1 Data for underlying data. GS, global signal.

might mediate observed time of day effects on GS fluctuation, we correlated 13 summary metrics (across domains of head, cardiac, and respiratory motion) to GS fluctuation (Fig 2A) and to time of day (Fig 2B) across participants for each run.

Some participants had to be excluded because of poor cardiac or respiratory recording quality (see Materials and methods). Therefore, we ended up with different subgroups of participants for the head motion, cardiac, and respiratory analyses. Consequently, we replicated the negative correlations between GS fluctuation and time of day for each subgroup (Fig 2A and 2B).

After correction for multiple comparisons (see Materials and methods), 12 of the 13 measures showed consistent correlations with GS fluctuation across all four runs, with the exception of temporal derivative of root mean square variance over voxels (DVARS) mean, which was significantly correlated with GS fluctuation in only one of the runs (Fig 2A). The strongest correlations with GS fluctuation were observed for standard deviation of respiratory variation (RV SD; average r = 0.55 across four runs), mean of respiratory variation (RV mean; average r = −0.40), standard deviation of heart rate (HR SD average r = 0.27), outlier frames % (average r = 0.26), and standard deviation of framewise displacement (FD SD; average r = 0.23).

However, when we correlated these same 13 measures to time of day (Fig 2B), only RV SD showed a significant correlation with time of day for all four runs (average r = −0.12) after correction for multiple comparisons (see Materials and methods). It is worth noting that the strength of this correlation was weaker than the correlation between GS fluctuation and time of day in the same subgroup of participants (average r = −0.18).

## Effects of time of day on GS fluctuation persist after correcting for respiratory variation

Given that both RV SD and GS fluctuation were robustly correlated with time of day (Fig 2B), we tested whether the effect of time of day on GS fluctuation remained significant after controlling for RV SD.

We found that RV SD was negatively correlated with time of day both between participants and within participants in both sessions in the subgroup of participants with high-quality respiratory data (Fig 3A and 3B and S3 Fig). Like GS fluctuation, RV SD was elevated in the second compared with the first run of each session (Fig 3C and S3 Fig).

We then repeated our original analyses (Fig 1) after regressing RV SD from GS fluctuation across participants. The negative correlation between time of day and GS fluctuation residual remained significant and only slightly attenuated in magnitude (Fig 3D, 3E and 3F, S3 and S4 Figs).

## Time of day effects on regional BOLD signal fluctuation are most prominent in sensory-motor regions

The significant negative correlation between time of day and GS fluctuation suggests that a substantial portion of the cortex might exhibit diurnal variation in activity during resting state. However, since the contribution of different regions to the GS is often found to be non-uniformly distributed across the brain [42–44], we investigated the effects of time of day on brain-wide regional BOLD signal fluctuation.

We considered a whole-brain parcellation consisting of 400 cortical regions (Fig 4A) [45] and 19 subcortical regions (Fig 4B) [46]. Regional BOLD signal fluctuation maps were derived for each participant by computing the standard deviation of the mean time series in each of the 419 regions of interest (ROIs) within each run. These run-level estimates were averaged within each session. We then correlated the regional BOLD signal fluctuation maps with time of day between participants for each session.

Regional BOLD signal fluctuation showed a negative association with time of day in sensory cortices, including visual cortex, somatomotor cortex, and the anterior cingulate for session 1

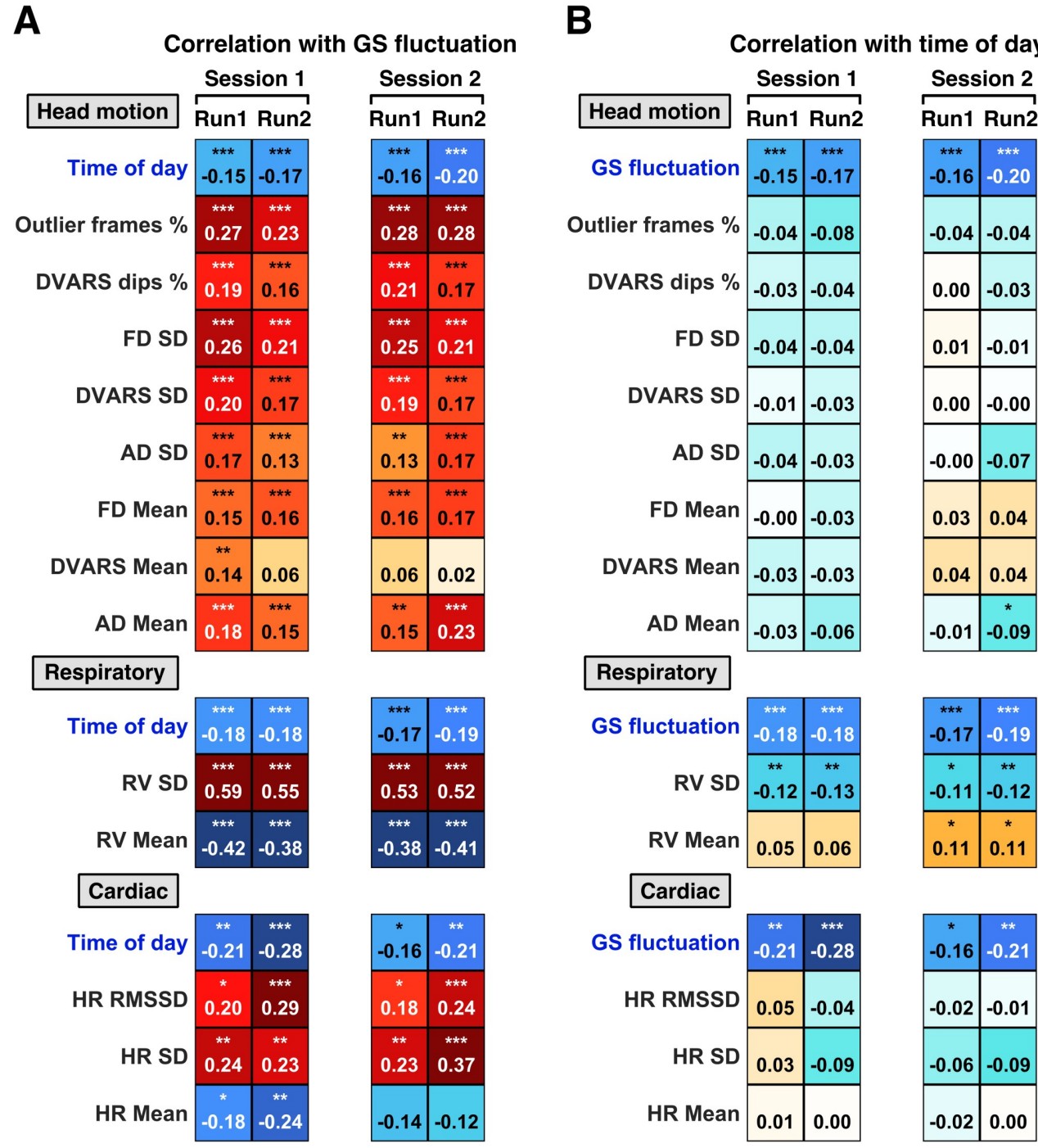

**Fig 2. Head motion, respiratory, and cardiac measures are strongly correlated with GS fluctuation, yet only respiratory variation shows association with time of day. (A)** Between-participant correlations of 13 run-level summary metrics with GS fluctuation. **(B)** Between-participant correlations of 13 run-level summary metrics with time of day. Because of exclusion of individuals with poor physiological data quality, different subgroups of participants were used for analyses of head motion (session 1: $N = 942$, session 2: $N = 869$), respiratory (session 1: $N = 741$, session 2: $N = 668$), and cardiac measures (session 1: $N = 273$, session 2: $N = 272$). Correlation between GS fluctuation and time of day was repeated in each subgroup. Numbers denote Pearson correlation coefficients. Stars indicate significant correlations following FDR correction (*$q < 0.05$; **$q < 0.01$; ***$q < 0.001$). See S2 Data for underlying data. AD SD, standard deviation of absolute displacement; DVARS SD, standard deviation of the temporal derivative of root mean square variance over voxels; FD SD, standard deviation of framewise displacement; FDR, false discovery rate; GS, global signal; HR RMSSD, root mean of the successive differences of heart rate; HR SD, standard deviation of instantaneous heart rate; RV SD, standard deviation of respiratory variation.

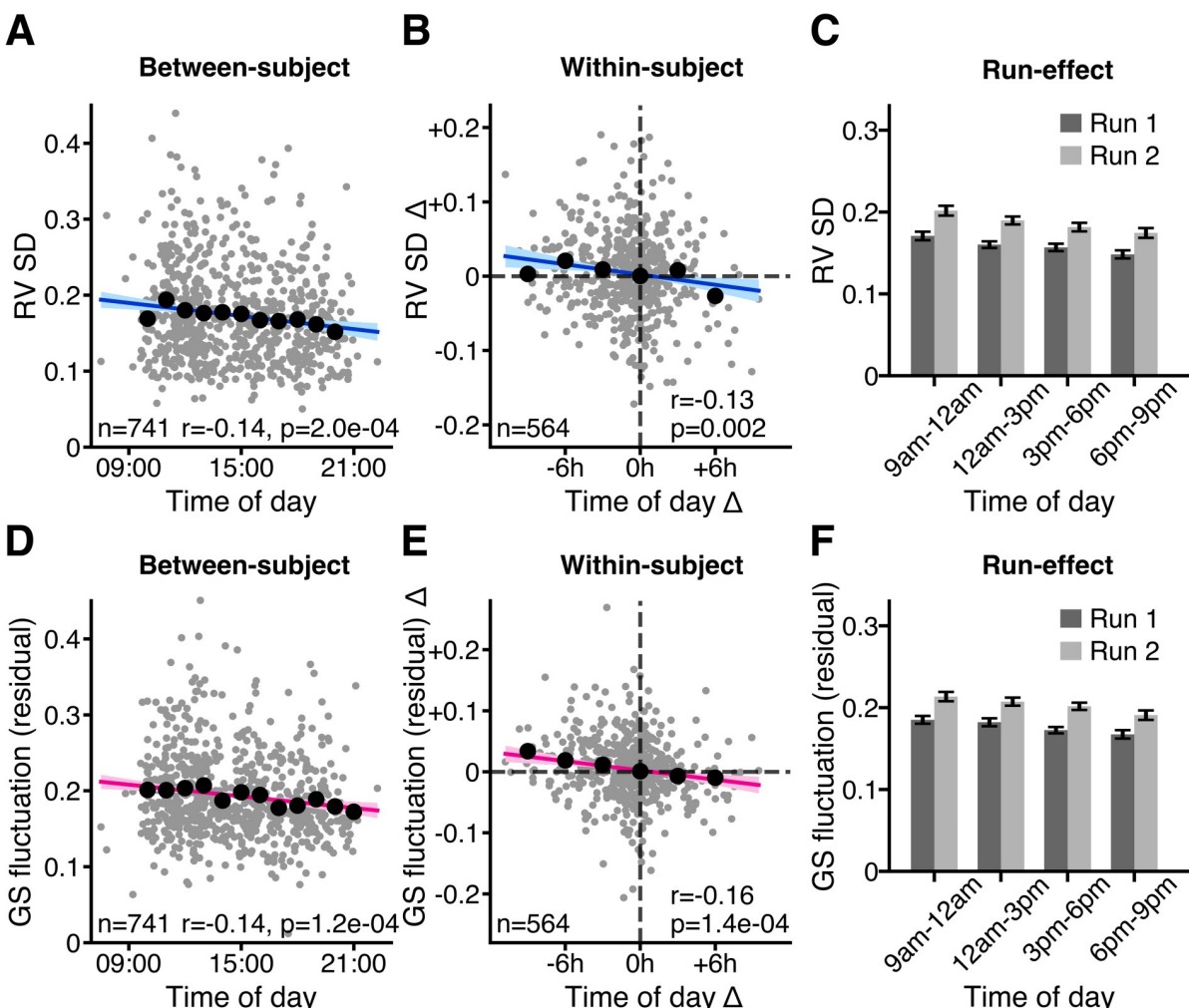

**Fig 3. Negative association between time of day and GS fluctuation remains significant after controlling for respiratory variation. (A)** Between-participant variation, **(B)** within-participant variation, and **(C)** run effects of RV SD as a function of time of day. **(D)** Between-participant variation, **(E)** within-participant variation, and **(F)** run effects of GS fluctuation residual (after regressing RV SD) as a function of time of day. Same as in Fig 1, within-participant effects were computed by taking the difference (Δ) for each variable between session 2 and session 1. Grey dots denote individual participants. Confidence interval is shown in light blue or in light pink. Black dots denote mean of GS fluctuation in hourly (**A, D**) or 3-hourly (**B, E**) time windows. R values denote Pearson correlation coefficients. *p*-Values were derived from 100,000 permutations while keeping family structure intact [40]. This figure shows the results for session 1 (see S3 Fig for session 2). See S1 Data for underlying data. GS, global signal; RV SD, standard deviation of respiratory variation.

(Fig 5A). The impact of time of day was generally stronger and more widely distributed for session 2, though again, strong effects were observed in visual and somatomotor regions.

Similarly, subcortical regions exhibited either no significant association or a significant negative association between time of day and regional BOLD signal fluctuation (Fig 5B). The strongest effects across the subcortex were found in the bilateral putamen and bilateral thalamus during session 1, effects that were replicated in the right hemisphere during session 2 (Fig 5B).

## Time of day is associated with widespread reductions in RSFC

There is significant interest in relating RSFC to individual differences in stable traits [37,48–50], an endeavour that is hindered by unaccounted variation in the behavioural or

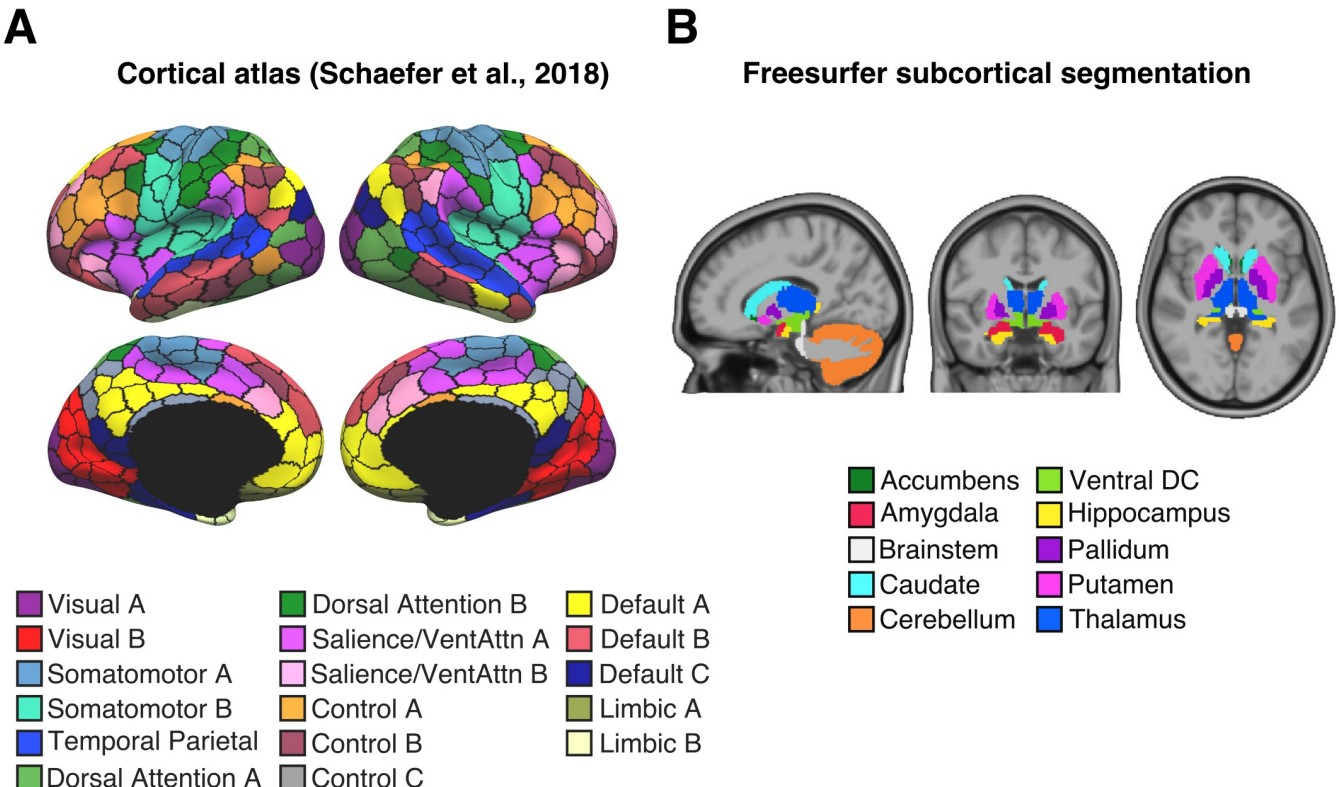

**Fig 4. Four hundred nineteen ROIs. (A)** Four hundred–area cortical parcellation in fs_LR surface space [45]. Parcel colours correspond to 17 large-scale networks [47]. *Image reproduced under a CC BY 4.0 license, credit: https://doi.org/10.6084/m9.figshare.10062482.v1* **(B)** Nineteen subcortical regions defined in participant-level volume space [46]. *Image reproduced under a CC BY 4.0 license, credit: https://doi.org/10.6084/m9.figshare.10063016.v1.* DC, diencephalon; ROI, region of interest; VentAttn, ventral attention.

physiological states of participants. This motivated us to assess whether variation in time of day of scans had a systematic effect on estimates of RSFC across participants.

Like before, we computed a 419 × 419 correlation matrix for each participant, reflecting RSFC between 419 ROIs spanning the cortex and subcortex (Fig 4), using the ICA-FIX denoised data from each run. Then, session-level RSFC matrices were derived by averaging the two run-level RSFC matrices within each session. Finally, we computed the correlation between time of day and the RSFC of each ROI pair across participants for each session. Results were visualised (Fig 6) both at the region level (lower triangular) and at the network level (upper triangular). Network-level results were computed by averaging Pearson r values across ROI pairs assigned to the same 17 large-scale networks defined in previous work [45,47].

Time of day exhibited a negative correlation with RSFC throughout the brain (Fig 6A). Although the magnitude of time of day effects varied across networks and regions, this variation was highly consistent between the two sessions (see S5 Fig for session 2). Prominent time of day effects were observed within somatomotor and visual networks and between these two and dorsal attention and default C network regions. In contrast, within-network RSFC in default B, control A and B, and limbic had minimal or no association with time of day.

To illustrate the magnitude of these effects, we also computed the correlation between RSFC and fluid intelligence. Fluid intelligence was chosen because it is a widely studied measure amongst resting-fMRI studies of brain-behavioural associations [50] and is one of the

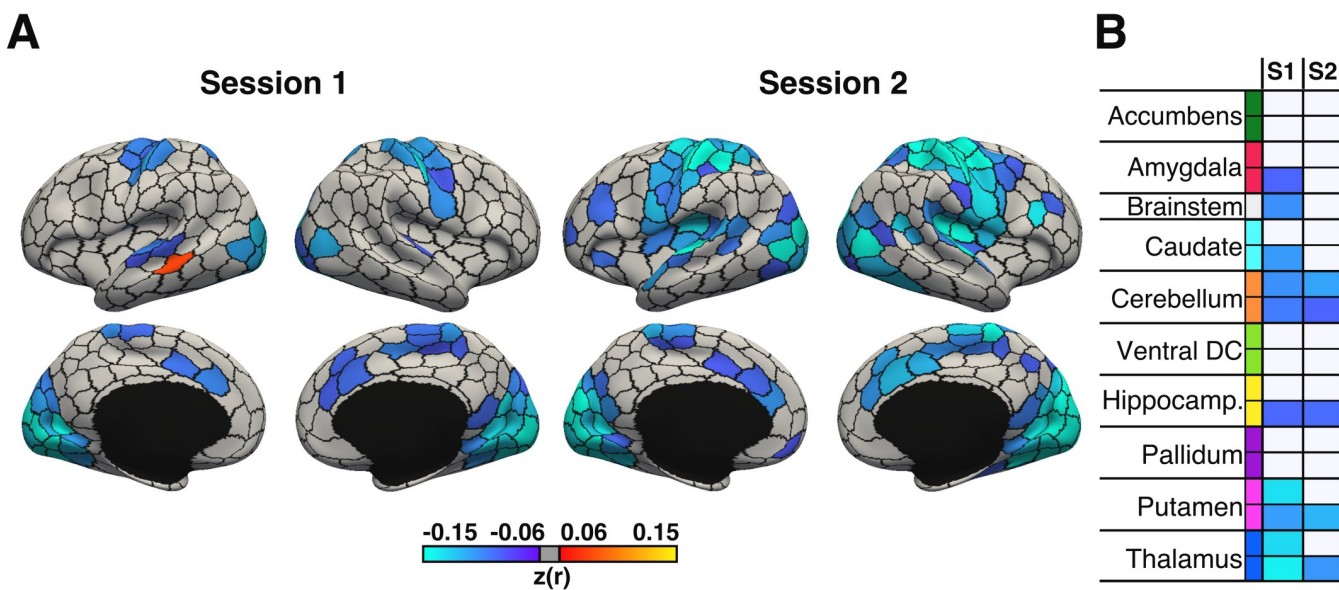

**Fig 5. BOLD signal fluctuation is negatively correlated with time of day across cortical and subcortical regions during session 1** (*n* = 942) **and session 2** (*n* = 869). Between-participant correlations between time of day and BOLD signal fluctuation for **(A)** 400 cortical and **(B)** 19 subcortical regions of interest. With the exception of brainstem, all subcortical regions are bilateral and presented as left-to-right hemisphere pairs (top to bottom). Colours (cool–warm) denote brain regions with significant Pearson r coefficients (q < 0.05, FDR-corrected), whereas nonsignificant regions are shown in grey. *p*-Values were derived from 100,000 permutations while keeping family structure intact [40]. BOLD, blood oxygen level–dependent; ventral DC, diencephalon; FDR, false discovery rate; Hippocamp., hippocampus; S1, session 1; S2, session 2.

behavioural measures best predicted by resting-state fMRI [37,38,52]. The correlation between time of day and RSFC was visibly and quantitatively stronger (Fig 6A; median absolute z = 0.13) than the correlation between fluid intelligence and RSFC (Fig 6B; median absolute z = 0.07) in session 1. This finding was replicated in session 2 (S5 Fig).

## GS regression weakens time of day effects but introduces novel effects in specific networks

GS regression (GSR) is a widely used denoising method effective at removing widely distributed (e.g., respiratory) artefacts from resting-state fMRI data [37,42,53–58]. However, GSR remains highly controversial because it might discard neural signals (potentially related to arousal) and introduce biases into the data [59–62].

Since we identified a robust association between time of day and GS fluctuation, we hypothesised that GSR might diminish time of day effects on regional BOLD signal fluctuation and RSFC. Thus, we repeated our original analyses of regional BOLD signal fluctuation and RSFC, but this time using time series that had undergone GSR (see Materials and methods).

GSR reduced the number of regions showing significant negative association between time of day and BOLD signal fluctuations (S6 Fig). However, GSR also introduced positive correlations in several regions that previously showed no significant association with time of day (S6 Fig).

GSR also resulted in an overall reduction of time of day effects on RSFC (S7 Fig; median absolute z values in session 1: z = 0.04, session 2: z = 0.04) relative to our original analyses without GSR (Figs 6A and S7; median absolute z values in session 1: z = 0.12, session 2: z = 0.13). Nevertheless, the effect of time of day on RSFC remained significant in both sessions following GSR as assessed by network-based statistics (false discovery rate [FDR]-corrected at q < 0.05).

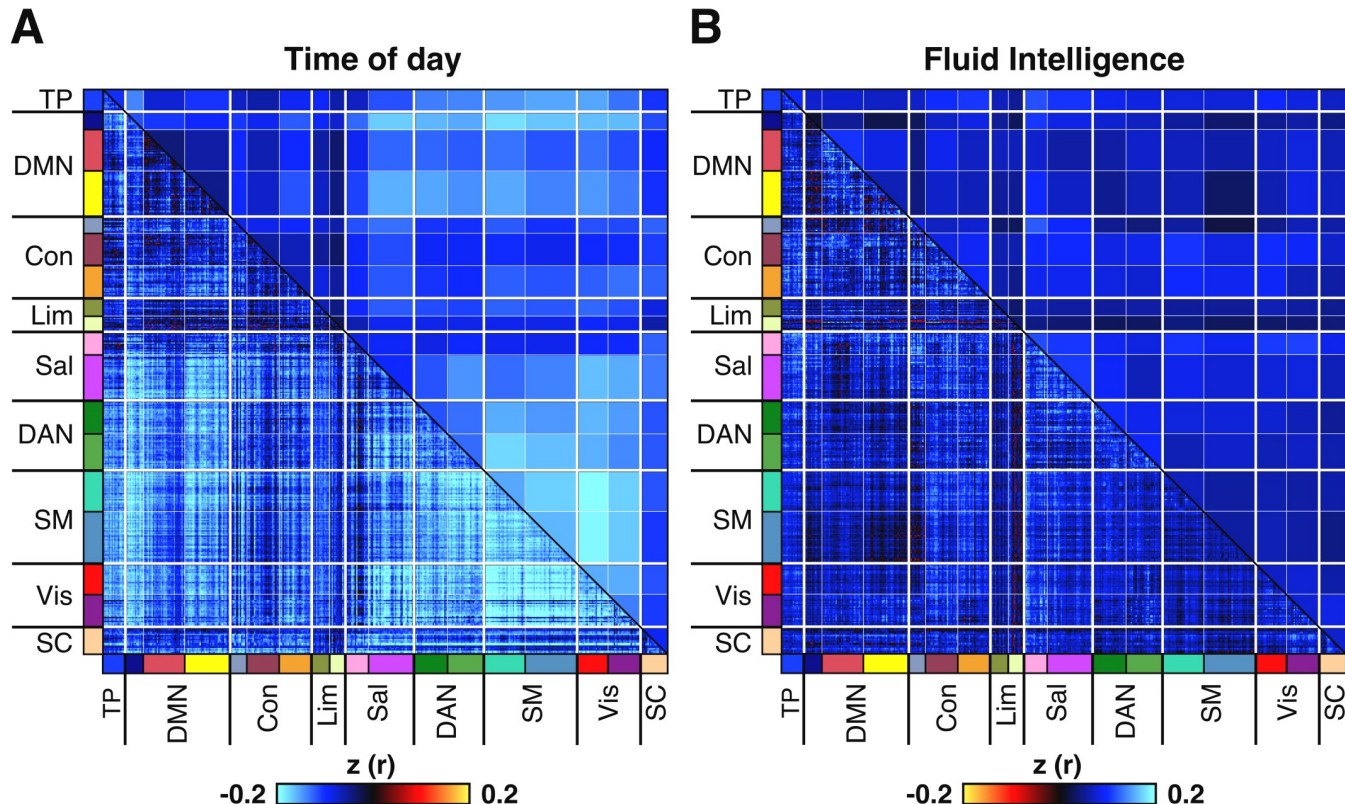

**Fig 6. RSFC is negatively correlated with time of day across participants, with a magnitude surpassing the strength of correlation between fluid intelligence and RSFC.** Results shown are for session 1 (see S5 Fig for session 2). **(A)** Correlation between time of day and RSFC across participants. **(B)** Correlation between fluid intelligence and RSFC across participants (with inverted colour scale to facilitate visual comparison with time of day effects). Colours in lower triangular of correlation matrix denote Pearson correlation coefficients. Colours in the upper triangular denote r values from the lower triangular averaged within network pairs. Colours on label axes denote correspondence of 419 regions to 17 large-scale cortical networks and to SC. Association between time of day and RSFC was significant in both sessions as assessed by network-based statistics (FDR-corrected at q < 0.05), whereas association between fluid intelligence and RSFC association was only significant in session 1 (FDR-corrected at q < 0.05). See Materials and methods for details of network-based statistics [51]. Fluid intelligence was chosen because it is a widely studied measure amongst resting-fMRI studies of brain-behavioural associations [50] and is one of the behavioural measures that is best predicted by resting fMRI [37,38,52]. Con, control network; DAN, dorsal attention network; DMN, default mode network; FDR, false discovery rate; fMRI, functional MRI; Lim, limbic network; RSFC, resting-state functional connectivity; Sal, salience network; SC, subcortical network; SM, somatomotor network; TP, temporal parietal network; Vis, visual network.

Furthermore, GSR also amplified the association between time of day and RSFC in certain large-scale circuits, e.g., somatomotor A–control and somatomotor A–salience B (S7 Fig).

## Control analyses

Given that some participants exhibited large GS fluctuation and RV SD, we performed several control analyses. We first visualised the standard error of the windowed means, which suggested similar spread of GS fluctuation across different times of day in between-participant analyses (S8 Fig). In within-participant analyses, the distribution was slightly more dissimilar across different times of day, with early and later times exhibiting greater standard error likely due to the availability of fewer data points than around midday (S8 Fig).

We also considered the possibility that the observed linear fit between GS fluctuation and time of day could have been influenced by heteroscedasticity of the data. Therefore, we carried out two additional types of regression in addition to ordinary least squares (OLS), which do not assume homoscedasticity: robust regression and quantile regression. We found the line of best fit to be highly similar across all three types of regression as shown in S9 Fig.

# Discussion

## Summary

Despite widespread evidence from animal [6–8] and human studies [9,16,17,19–23] that brain activity exhibits diurnal variation, most fMRI studies rarely consider the potential effects of time of day. In this study, we examined the impact of time of day on various measures of resting-state brain activity in over 900 participants from the HCP dataset, scanned between 8 AM and 10 PM on two sessions. We observed time of day–dependent reductions in the magnitude of GS fluctuation, regional BOLD signal fluctuation, and reductions in whole-brain RSFC throughout the course of the day. We also demonstrated that the magnitude of time of day–associated whole-brain RSFC decrease was larger than the association between RSFC and fluid intelligence.

## Time of day and arousal have opposing effects on GS fluctuation at distinct timescales

Surprisingly, the directionality of our main findings was in direct contrast to our hypothesis, which was motivated by models of circadian fluctuations in arousal levels [26,32–34]. We hypothesised that the magnitude of GS fluctuation, a measure associated with drowsy states [25,27,30], would be lowest in the late morning and early evening hours, when levels of arousal are typically at peak levels, and highest in the midafternoon, when levels of arousal often dip. Instead, we observed the highest levels of GS fluctuation in the late morning hours, followed by a gradual reduction of GS fluctuation across participants throughout the day until the late evening hours. Although the HCP resting-state scan protocol did not involve concurrent assessment of arousal (e.g., [29,63,64]), it is unlikely that arousal levels would be lowest in the late morning and monotonically rise throughout the entire day.

However, we did observe a robust increase in GS fluctuation in the second run of each session. Similar within-session variation in the magnitude of BOLD signal fluctuation has been previously reported in the HCP [41] and in other datasets [65]. Furthermore, simultaneous resting-state fMRI and polysomnographic electroencephalography (EEG) recordings have shown that participants exhibit an increased propensity to fall asleep with longer duration of time spent in the scanner [65]. These studies, as well as a wealth of earlier work linking higher regional BOLD signal [66–70] and GS fluctuation [25,27,30] to low arousal states, suggest that this phenomenon most likely reflects a drop in arousal as participants become progressively drowsier while lying in the scanner. Interestingly, we found that this between-run increase in GS fluctuation levels was present throughout different times of day in both sessions. Thus, we observed a run-level increase in GS fluctuation on the scale of minutes that was superimposed on a slower downward trend in GS fluctuation on the scale of hours.

## Potential roles of homeostatic mechanisms and CBF changes

What mechanism could account for the observed time of day–associated reductions in GS fluctuation and RSFC if not circadian variation in arousal levels as we had hypothesised originally? Animal studies have suggested that wakefulness and sleep might be respectively associated with homeostatic upscaling and downscaling of synaptic strength [6–8,71]. Building on this work, human studies have found that cumulative wakefulness is associated with a gradual rise in cortical excitability, as measured by transcranial magnetic stimulation–evoked EEG potentials [9,10] and motor-evoked potentials [10]. Interestingly, in one of these studies, the increase in cortical excitability was detectible even prior to sleep deprivation and was not associated with concomitant changes in behavioural indices of arousal [10]. However, it is not

directly apparent how a diurnal increase in synaptic strength or cortical excitability would result in the time of day–associated reductions in GS fluctuation and RSFC that we observed in our study.

Based on the premise that synaptic strengthening has a significant metabolic cost [72], the synaptic homeostasis hypothesis [71] predicts an increase in cerebral oxygen and glucose consumption and, as a result, increased CBF with cumulative wakefulness [73]. Indeed, some studies have reported elevated CBF in the evening relative to that in the morning [13,15], though not all [14,23].

Interestingly, the largest of these studies [15] reported regional CBF increases related to duration of wakefulness in the same regions where we observed the strongest time of day–associated reductions in resting-state BOLD signal fluctuation and RSFC (e.g., somatomotor and visual cortex). These same regions have also been shown to exhibit reduced task-related BOLD signal activation as a function of duration of wakefulness [17]. This is of particular relevance, since elevation of baseline CBF levels via experimentally induced hypercapnia (inhalation of $CO_2$-enriched air) is known to reduce the dynamic range of resting-state BOLD fluctuation and RSFC [74–76]. Thus, a time of day–associated increase in baseline CBF would present a possible mechanism that could account for our observed reductions in GS fluctuation and RSFC. Nevertheless, the jury is still out on whether these effects are indeed the consequence of homeostatic upscaling of synaptic strength or another mechanism.

## Respiratory variation also shows time of day modulation but does not account for changes in GS fluctuation

Since respiration is a potent modulator of BOLD signal fluctuation [57,77,78], we considered whether diurnal changes in respiratory variation existed and whether these could account for the time of day–associated reductions in GS fluctuation. Indeed, we observed a significant negative correlation between the RV SD and time of day. Thus, individuals scanned in the earlier hours of the day exhibited more dynamic breathing patterns (potentially more sighs, yawns, pauses, or variation in breathing rate or depth) than those scanned in the evening. The mechanisms behind this are unclear, although earlier studies have proposed that breathing stability might be impaired in the morning hours, as suggested by augmented ventilatory response to hypoxia [79] and hypercapnia challenges [80].

Variation in respiratory activity, whether spontaneous [42,57] or instructed [78,81], is known to introduce variance into the BOLD signal via $CO_2$-associated fluctuations in CBF [77]. Indeed, we observed a strong positive correlation between RV SD and GS fluctuation, replicating earlier findings from other datasets [42,57]. Furthermore, we found that the between-run elevation of GS fluctuation was also accompanied by a between-run elevation of RV SD. Given the strong statistical relationship between the two measures, as well as a plausible mechanistic link [77], we thought that respiratory variation might account for the effect of time of day on GS fluctuation. However, the negative association between time of day and GS fluctuation remained significant and only slightly diminished following regression of RV SD from GS fluctuation. Thus, respiratory variation only appeared to account for a small component of the observed diurnal downward shift in GS fluctuation.

Admittedly, linear regression of RV SD from GS fluctuation might miss potential nonlinear effects of respiration on GS fluctuation. Modelling of respiratory influences on BOLD signal fluctuations is nontrivial, as respiratory response functions often fail to capture the impact of pauses and subtle variations in rate and depth of breathing [42,78]. Recently, it has also been shown that resting-state BOLD signal fluctuations that track respiratory dynamics exhibit spatially heterogeneous time courses (lags and shape) throughout the brain [82]. Thus, we cannot

fully rule out respiratory mechanisms underpinning our observed time of day effects on BOLD signal fluctuation.

## Spatial distribution of time of day effects

Time of day was negatively correlated with regional BOLD signal fluctuation and RSFC across many brain regions, with the strongest effects apparent in visual and somatomotor cortex. These findings are consistent with several observations (though not all) from previous smaller-scale studies [19,20,22].

One of the first resting-state fMRI studies to look at time of day effects found that within-network strength and spatial extent of the somatomotor network decreased as a function of time elapsed since participants' middle hour of sleep, whereas executive control network connectivity was found to be relatively stable over time [22]. Later, Jiang and colleagues showed reduced amplitude of low-frequency fluctuations and homogeneity in sensory-motor regions in evening versus morning scans [20]. More recently, Cordani and colleagues reported diurnal reductions in sensory-motor regions during morning and evening twilight hours, while also noting that reductions of BOLD signal fluctuation in visual cortex were associated with improved performance on a visual detection task [19]. To our knowledge, ours is the first study to report a cumulative reduction of resting-state GS fluctuation and regional BOLD signal fluctuation.

A number of studies outside the context of investigating time of day effects have shown that sensory-motor regions exhibit particularly high levels of intraparticipant variation in BOLD signal fluctuation and RSFC across the brain [38,41,83,84], an observation that is typically attributed to modulations of arousal state [41,62,65,85,86].

However, Power and colleagues have cautioned that since regressors that account for respiratory signals also exhibit a strong loading on these same regions, the high levels of variability in these regions could reflect respiratory artefacts rather than neural activity related to arousal [87]. Furthermore, they have drawn attention to the fact that there is a high prevalence of isolated deep breaths during resting-state scans [88]. These large breaths produce BOLD signal modulations spanning 30–40 seconds that are far slower than the standard evoked response function used in task fMRI to model neural activity [88,89].

The challenge of distinguishing neural fluctuations from potential respiratory artefacts is made difficult by the fact the respiratory and neural dynamics are unlikely to be independent even without the presence of artefacts. For example, a recent study found that natural fluctuations in arterial $CO_2$ due to respiration appeared to track oscillatory power in multiple frequency bands as measured by magnetoencephalography (MEG) [90].

Since electrophysiological techniques, such as MEG and EEG, do not rely on neurovascular coupling, they provide a more direct measurement of neural activity without fMRI's susceptibility to respiratory artefacts. Indeed, studies employing electrophysiology in tandem with fMRI or functional ultrasound imaging have provided many examples in which global hemodynamic fluctuations coincide with changes in neural activity as indexed by local field potentials (LFPs) [27,29,91–93].

Chang and colleagues reported that the global fMRI signal showed a strong temporal correlation with an LFP-based index of arousal in macaques [29]. In another study, Liu and colleagues found that global peaks observed in fMRI signal were associated with spectral shifts in LFP in macaques and that peaks defined in the same manner exhibited a visual-sensorimotor coactivation pattern in humans [86]. More recently, Bergel and colleagues used ultrasound imaging in rats to show that brain-wide vascular surges lasting between 5 and 30 seconds during REM sleep were preceded by sustained activity in the theta, mid-gamma, and high-gamma

LFP bands [93]. Importantly, Turchi and colleagues demonstrated that pharmacological inactivation of basal forebrain nuclei selectively decreased global fMRI signal fluctuation ipsilaterally to the injection site even after statistically accounting for end-tidal $CO_2$ partial pressure ($pCO_2$) fluctuation and head motion [94].

## Head motion, heart rate variability, and cerebrovascular reactivity

Recent work has shown that substantial individual variation in GS fluctuation [42,57] and RSFC [94] can be explained by summary metrics derived from recordings of physiological variables, even after thorough preprocessing of the data. Our findings corroborate this view to the extent that we also observed strong correlations between GS fluctuation and a wide range of summary metrics across domains of respiration, head motion, and cardiac measures. Yet, when we correlated these same metrics with time of day, only RV SD was consistently correlated with time of day across the four runs in two sessions. Thus, individual differences in head motion and cardiac measures, although contributing to variation in GS fluctuation, did not account for the effects of time of day on resting brain activity.

Can time of day–related variation in cerebrovascular reactivity explain our findings, whereby the same level of respiratory or neural activity might produce a stronger modulation of the BOLD signal at different times of day? Indeed, cerebrovascular reactivity has been shown to be lower in the morning relative to evening hours, as measured by middle cerebral artery velocity and cerebral oxygenation following breath holds or hypercapnia challenges [80,95,96]. However, this account is not consistent with our observations of increased BOLD signal fluctuation and RSFC in the morning compared to evening, since lower cerebrovascular reactivity is typically associated with lower, not higher, BOLD signal fluctuation and RSFC [97].

## Implications for GSR

Our results suggest that although GSR does not fully eliminate time of day effects on RSFC, it does lead to an overall reduction in time of day effects across the brain. Interestingly, the negative association between time of day and BOLD signal fluctuation remained strongly preserved in the visual cortex, in both sessions, even following GSR. This suggests that the visual cortex might be particularly susceptible to diurnal variation, an idea that is consistent with recent task-activation studies [17,19].

The validity of GSR has been extensively debated in the field of resting-state fMRI [61]. Proponents of GSR have emphasised its ability to remove spatially widespread respiratory artefacts [42,55,57,87], whereas others have raised concerns that this may entail removal of neural signal [44,62,98,99] or the introduction of spurious relationships into the data [59,60,100]. Our control analyses suggested that the effect of time of day on GS fluctuation could not be accounted for by individual differences in measures that might suggest nonneural sources of variance, e.g., head motion, respiratory, and cardiac measures. Thus, assuming that our results indeed reflect diurnal variation in large-scale spontaneous brain activity, then the current study could represent an example of where GSR might be discarding signal of neural origin.

Whether GSR is preferable or not, however, still might vary according to the dataset at hand (e.g., extent of global artefacts in the data) and on the specific aims of the study. For example, it is possible that time of day effects on resting brain activity might simultaneously manifest both as globally distributed decreases in RSFC and as more subtle, localized increases of RSFC between a few select regions or networks. These latter effects, however, might only become apparent once the GS is removed from the data, for example, as was the case with the

positive correlations between time of day and somatomotor–control network RSFC in the current study (S7 Fig).

Alternatively, the positive correlations observed between time of day and somatomotor–control network RSFC might be artefacts introduced by GSR. Indeed, several studies have shown using simulations that GSR introduces negative connectivity, biases short- and long-distance connectivity, and can lead to spurious group differences [59,60,100]. However, others have cautioned that these simulations may not generalise well to empirical data in which the number of independent signals is substantially larger or in which there is a large shared artefactual signal among brain regions [55,101].

There has been substantial interest in the field to pioneer alternatives to GSR [57,62,102, 103], with the aim of removing putative global artefacts, without some of the purported drawbacks of GSR. Nevertheless, the validity and utility of various approaches for dealing with GS remains a topic of debate and enquiry [37,61,62,87,99,104].

### Relationship to studies of diurnal variation in brain structure

Time of day effects have been observed not only in studies of brain activity but also in studies of brain structure [105–110]. Notably, some studies have reported morning-to-evening reductions in total brain and/or grey matter volume at rates comparable to age-related annual atrophy [107,110]. Hydration levels have been mentioned as a possible mechanism, given that hydration has been shown to modulate measurements of total brain volumes in some studies [111,112], though not all [110]. Dehydration could also potentially impact BOLD signal fluctuation by reducing the amplitude of brain (or cerebrospinal fluid) pulsation or the dynamic range of cardiac and respiratory activity. Unfortunately, fluid intake of participants was not tracked in the HCP protocol.

### Methodological limitations

Since our main finding of a time of day–associated reduction in GS fluctuation was derived based on group-level analysis, we cannot confidently infer what proportion of individuals actually exhibited this tendency within our sample. It is possible that for a subset of our sample, there was no effect of time of day on GS fluctuation, or there was an effect with a different pattern than the one observed across the group.

The HCP protocol did not involve assessment of arousal state during the resting-state scans. Methods typically used in the field include tracking of eyelid closures [63], pupillometry [85], EEG [27], polysomnography [65], or cognitive vigilance [17]. Such data would have helped interpret the unexpected directionality of our findings by providing a more direct measure of arousal state, e.g., changes in the frequency of spontaneous eye closures or duration of in-scanner sleep. Similarly, measurement of spontaneous fluctuations in $pCO_2$ levels via a nasal cannula during the resting-state scans [77] would have enabled a more comprehensive characterisation of the effects of time of day on respiration and the extent to which these effects can potentially account for diurnal variation in GS fluctuation.

Our ability to detect an association between time of day and GS fluctuation across participants had as much to do with a sufficiently low interparticipant variation in GS fluctuation as with a sufficiently large effect of time of day on GS fluctuation. For example, our inability to detect time of day effects on head motion or heart rate variability measures might have been due to greater baseline interindividual variation of these measures relative to the magnitude of the impact of time of day. Thus, we cannot fully rule out the existence of time of day effects for these other measures.

There are several unaccounted variables, which could have potentially impacted the observed diurnal variation in resting brain activity. In the current study, time of day likely correlates with the hours of scan usage on a given day; thus, we cannot rule out the presence of scanner use–related hardware artefacts. However, we note that such hardware artefacts are unlikely to produce time of day effects with a spatial specificity to visual and somatomotor cortices. Although caffeine consumption was not monitored, increased morning caffeine consumption would not account for elevated GS fluctuation, since caffeine is known to reduce GS fluctuation [31]. Other examples of unaccounted variables include meal timing, meal size and composition, fluid intake, bedtime on previous night, wake-up time on the day of study, timing of commute, weather, and light levels. Finally, the experience itself of taking part in an extensive study and of being scanned could have resulted in deviations from typical diurnal fluctuations of arousal. For example, participants scanned in the morning versus evening could have experienced differential levels of arousal anxiety or stress due to a buildup of anticipation to being scanned.

## Recommendations

In this paper, we have shown systematic time of day effects in the HCP dataset on resting-state fMRI measures that were greater in magnitude to behavioural associations (e.g., fluid intelligence), consistent in both between- and within-participant analyses, present following regression of respiratory variation, and replicable across two sessions. Leaving aside the important challenge of understanding their underlying mechanisms, another important question to address is what type of studies in the field are most likely to be affected by time of day and to what extent.

Large-scale studies such as the HCP are particularly susceptible to time of day effects because they are more likely to have individuals scanned over a wide range of times in order to facilitate data collection. Although we found time of day effects to be modest in absolute terms, explaining less than 4% of the variance in GS fluctuation or RSFC, these were still comparable or stronger than most behavioural–RSFC associations reported in fMRI studies. Thus, group-level correction for time of day could potentially help avoid masking out or introducing spurious behavioural–RSFC associations in other similar large-scale studies with varying times of day of scans.

In contrast, most small-scale studies are unlikely to be affected, as these tend to scan participants in fixed time slots. Even studies in which participants are scanned several hours apart are not likely to be drastically impacted based on the fact that it takes around 9 hours for time of day effects to reach a similar magnitude as run effects on GS fluctuation (see Fig 1E). Notable exceptions are longitudinal studies that seek to examine neural changes associated with same-day skill acquisition, as recently shown by Steel and colleagues [21]. In general, all studies should avoid a situation in which there is nonrandom assignment of an experimental group or condition to a specific time of day. Group-level correction for time of day is unlikely to be useful in small-scale studies, as they may lack statistical power to reliably estimate time of day effects.

We recommend reporting time of day of fMRI scans and other experimental protocols and measurements. Even if all participants are scanned at the same time slot within a particular study, reporting time of day could potentially account for between-study variation in results and failed replications. Meta-analyses could also then be leveraged to explore how time of day affects various regions, networks, and tasks across different domains of the literature.

## Future work

There are many avenues to extend the current study. For example, it will be interesting to explore whether the same effects can be seen during task fMRI. In addition, Glasser and

colleagues [62] proposed the use of temporal ICA to decompose the fMRI data into multiple components, some of which appear to reflect "global" artefacts, which can then be more selectively removed, and some of which may relate to neural signals for arousal or eyes-open versus eyes-closed states. It would be interesting to investigate how these distinct "global" components might relate to time of day. Furthermore, some "global" components are present during only resting fMRI but not task fMRI [62]. Thus, some of the effects we observe in this study might not appear in task fMRI.

## Materials and methods

### Ethics statement

This paper utilised data collected for the HCP [113]. The scanning protocol, participant recruitment procedures, and informed written consent forms, including consent to share deidentified data, were approved by the Washington University institutional review board [113]. Our use of the HCP data for this research was carried out with local institutional review board approval at the National University of Singapore (N-17-056).

### Data overview

**Participant characteristics.** All data analysed in this manuscript are from the S1200 release of the HCP dataset [24,35], which was drawn from a population of young, healthy twins and siblings. Participants underwent a range of neuroimaging protocols, including resting-state fMRI, which was collected over two sessions, with each session split into two consecutive runs. We only considered individuals who had available resting-state fMRI data from at least one of the two sessions, acquired with both left-to-right and right-to-left phase encoding. We excluded from analysis any session in which the participant exhibited excessive head motion in more than 50% of the acquired frames in either run. Following exclusion of excessive movers, we retained 942 individuals from session 1 and 869 individuals from session 2. We conducted separate, parallel analyses for each session, with the exception of the within-participant correlations, which required analyses of participants with data from both sessions ($N$ = 831). Additional exclusions were applied selectively for analyses of cardiac and respiratory data based on the quality of recordings and for analyses of RSFC because a handful of participants had missing fluid intelligence scores (see below).

**Image acquisition.** Imaging took place at Washington University on a custom-made Siemens 3T Skyra scanner. Four runs of resting-state fMRI data were collected over two sessions (two on each session), which for the majority took place on separate days. During each run, 1,200 frames were acquired using a multiband sequence at 2-mm isotropic resolution with a TR of 0.72 seconds over the span of 14.4 minutes. Participants were instructed to maintain fixation on a bright crosshair presented on a dark background in a darkened scanning room. The resting-state sequences were collected at the start of each scanning session, preceded only by brief field-map and BIAS acquisitions. Structural images were acquired at 0.7-mm isotropic resolution. Further details of the data acquisition parameters can be found in previous publications [24,35,114].

**Preprocessing of structural images.** T1 structural images were preprocessed using a custom pipeline that utilised Freesurfer [46] along with other software and algorithms (e.g., [115–118]). Preprocessing steps included brain extraction, participant-level volumetric segmentation of subcortical regions, cortical surface reconstruction, and registration of each participant's cortical surface mesh to a common spherical coordinate space [119,120]. For a more detailed description of the structural preprocessing pipeline, see [114].

**Preprocessing of resting-state fMRI.** The HCP release includes resting-state fMRI data at two stages of preprocessing: minimally preprocessed and spatial ICA-FIX cleaned data [114]. The minimally preprocessed data had undergone removal of spatial distortion, motion correction via volume realignment, registration to the structural image, bias-field correction, 4D image intensity normalization by a global mean, brain masking, volume-to-surface projection, multimodal interparticipant alignment of the cortical surface data [118], and 2-mm (FWHM) parcel- and surface-constrained smoothing. The resulting image is a CIFTI time series with a standard set of grayordinates at 2-mm average surface vertex and subcortical volume voxel spacing. The HCP release of the spatial ICA-FIX processed data (ICA-FIX) had undergone additional preprocessing steps following minimal preprocessing.

The data were first de-trended with a temporal high-pass filter, followed by regression of head-motion parameters, before undergoing denoising via spatial ICA-FIX [121,122].

**Statistical analyses.** Given the high prevalence of families in the dataset, we applied a method of permutation testing that respects the family structure of participants during shuffling [40]. $p$-Values were calculated from 100,000 permutations. FDR correction was applied at q < 0.05 to all 1,815 statistical tests conducted in this manuscript.

## Time of day effects on GS fluctuation

We assessed the relationship between time of day and GS fluctuation between participants in each session, followed by a within-participant (i.e., between-session) assessment of the same relationship. The between-participant association between time of day and GS fluctuation was computed for each session by first averaging time of day and GS values across the two runs for each participant, followed by computing a Pearson correlation coefficient between the two session-level average values. For the computation of within-participant correlation, we again averaged each measure across the two runs of each session for each participant and then used the within-participant difference (Δ) of each measure between the two sessions to compute a Pearson correlation. GS fluctuation was always plotted in units of BOLD percent signal change. This was computed by dividing each participant's GS fluctuation score by 10,000 and multiplying by 100, given that each resting-state fMRI run of each participant had originally undergone grand-mean scaling to a value of 10,000 during preprocessing.

## Assessment of nonneural contributions to time of day effects

**Head motion, cardiac, and respiratory summary metrics.** We also examined the impact of time of day on a range of head motion, respiratory, and cardiac variables, which have been highlighted in the literature as potential indicators of nonneural sources of variance. In each of the four runs, we calculated 13 participant-level summary metrics: percentage of frames that are outliers (outlier frames %), percentage of DVARS dips/peaks (DVARS dips %), standard deviation of framewise displacement (FD SD), mean of FD (FD mean), standard deviation of absolute displacement (AD SD), mean of absolute displacement (AD mean), standard deviation of DVARS (DVARS SD), DVARS mean, RV SD, RV mean, root mean of the successive differences of heart rate (HR RMSSD), standard deviation of instantaneous heart rate (HR SD), and mean of heart rate (mean HR).

Outlier frames were defined as those in which DVARS exceeded 75 or in which FD exceeded 0.2 mm, as well as frames one before and two after such outlier frames, and nonoutlier segments consisting of fewer than five consecutive frames. This definition is based on earlier work on head-motion censoring [55,123], and the specific parameters are consistent with our recent study using the HCP dataset [38]. FD was defined as the summed root-mean-square of the derivatives of six motion parameters (three rotational, three translational), AD was

defined as the summed root-mean-square of six motion parameters (three rotational, three translational), and DVARS was defined as the root-mean-square of voxelwise differentiated signal. FD, AD, and DVARS were all computed using the FSL function *fsl_motion_outliers* [124]. DVARS dips/peaks were defined for each participant as the percentage of frames, which deviated by at least 75 from the median DVARS value [62]. Based on prior work, we derived DVARS dips/peaks from unstructured noise time series [62]. Unstructured noise time series were computed by regressing each ICA-FIX signal component from the ICA-FIX denoised image of each participant. RV was defined as the standard deviation of the respiratory waveform within 5.76-second windows (equal to eight fMRI frames) centred on each fMRI frame [125]. Heart rate was calculated at each peak based on the preceding interbeat interval.

We correlated (Pearson r) each of the 13 summary metrics separately with both GS fluctuation and time of day across participants. Our intent for correlating them with GS fluctuation was to provide us with a positive control, as many of these measures have been shown to be associated with GS fluctuation in other data [42]. Significance level was calculated using 100,000 permutations, while respecting family structure of participants. We repeated the FDR correction at two additional thresholds, thus allowing us to denote the level of significance for each correlation in Fig 2 (*q < 0.05; **q < 0.01; ***q < 0.001).

**Cohorts for analysis of respiratory and cardiac measures.** All pulse and respiratory time series in the S1200 HCP dataset underwent visual inspection by the same person (CO). For a participant's data to be included in the analysis of a session, both runs had to pass quality control. Following quality control, a substantial proportion of the pulse data, and a somewhat lesser proportion of the respiratory data, were considered to be inadequate for analysis. Therefore, separate subgroups were used for analysis of head motion (session 1: $N = 942$; session 2: $N = 869$), respiratory (session 1: $N = 741$; session 2: $N = 668$), and cardiac summary metrics (session 1: $N = 273$; session 2: $N = 272$). The list of individuals who passed visual quality screening of their pulse and respiratory data is publicly available at the GitHub repository maintained by the Computational Brain Imaging Group (https://github.com/ThomasYeoLab/CBIG/tree/master/stable_projects/preprocessing/Orban2020_tod).

**Quality control of pulse data.** We used the pulse oximeter and respiratory waveform data that were acquired (at 400 Hz) during the resting-state fMRI. We applied automated peak detection on the pulse wave data (https://github.com/demotu/BMC/blob/master/functions/detect_peaks.py), followed by individual visual inspection of each peak, and manual correction when necessary. Runs containing peaks that could not be reliably identified because of possible motion or hardware artefacts were excluded. Rare isolated ectopic beats that were not followed by a compensatory pause were manually deleted but were not a reason for excluding an entire run. Some runs with clearly identifiable peaks were excluded if they contained extended segments that were deemed to be arrhythmic or artefactual in origin (S10 Fig). Our aim, in this study, was not to classify different types of arrhythmias in the HCP dataset but to exclude runs in which abnormal heart rhythms might inflate summary measures of heart rate variability, which under healthy sinus rhythm reflect the influence of the autonomic nervous system [126]. In a number of runs, participants exhibited a heart rate of exactly 48.0 beats per minute with close to 0 interbeat interval variability, which we interpreted as artefactual in origin. These runs could be easily identified as outliers on Poincaré plots or on heart rate versus heart rate variability scatter plots (S10 Fig) and were excluded from further analysis.

**Quality control of respiratory data.** The raw respiratory waveform data of each participant from each run was z-scored, low-pass filtered at 2 Hz, and visualised in its entirety, and then each run was manually checked for sufficient data quality. RV, our primary measure of interest, was defined as the standard deviation of the filtered respiratory waveform within 5.76-second sliding windows (equivalent to eight TRs) and plotted alongside the respiratory

waveform data. The respiratory waveform frequently contained trains of high-frequency arte-facts. However, in most cases, these artefacts did not lead to changes in RV, as RV indexes fluc-tuations on a slower timescale that is calibrated to measure breathing patterns. We found the low-pass filter to be effective at eliminating many of these high-frequency artefacts, without distorting RV in clean segments of data. Runs with poor signal quality, ambiguous waveforms, or sustained high-frequency artefacts that contaminated RV time series even after bandpass fil-tering were excluded from analysis.

## Time of day effects on regional BOLD signal fluctuation

To assess the spatial distribution of time of day effects across the brain, we computed the Pear-son r correlation between the time of day of each participant's scan and the corresponding regional BOLD signal fluctuation measured across 419 ROIs. Our ROIs consisted of 400 corti-cal areas from the Schaefer parcellation defined in the HCP's fs_LR space (Fig 4A) [45] and 19 subcortical regions defined in participant-level volumetric space using Freesurfer's structural segmentation algorithm (Fig 4B) [46]. Regional BOLD signal fluctuation maps were first defined in each run as the standard deviation of resting-state BOLD signal averaged within each cortical or subcortical area of interest. The resulting matrices ($N \times 419$) were averaged between runs within each session, which were then used to assess the relationship between time of day and regional BOLD signal fluctuation.

  $p$-Values were computed using 100,000 permutations and corrected for multiple compari-son using FDR correction (q < 0.05). Results relating to cortical and subcortical ROIs were visualised in separate formats. Time of day effects on cortical regions were visualised on an fs_LR cortical surface mesh via Freeview, a part of the Freesurfer suite [46]. Each cortical region was assigned a colour label (shown in Fig 4A) based on their correspondence to 17 large-scale networks [45,47] consisting of temporo-parietal, default (A,B,C), control (A,B,C), limbic (A,B), salience, ventral attention, dorsal attention (A,B), somatomotor (A,B), and visual (A,B) networks. The results relating to the subcortical ROIs were visualised in a heatmap. The 19 subcortical ROIs consisted of left and right hemisphere masks of the nucleus accumbens, amygdala, hippocampus, caudate, thalamus, putamen, pallidum, ventral diencephalon, and cerebellum and a single mask of the brain stem.

## Time of day effects on RSFC

We assessed the relationship between time of day and RSFC and between RSFC and fluid intel-ligence. Fluid intelligence was chosen because it is a widely studied measure amongst resting-fMRI studies of brain-behavioural associations [50] and is one of the behavioural measures that is best predicted by resting fMRI [37,38,52]. We computed a Pearson r correlation matrix ($419 \times 419$) for each run of each participant using the mean time series extracted from the 419 ROIs described above. The run-level correlation matrices were averaged within sessions to produce session-level estimates of RSFC for each participant. These matrices were then concatenated ($N \times 419 \times 419$) and correlated with time of day and with fluid intelligence at each ROI to ROI pair ($N \times 419 \times 419$). We applied network-based statistics [51], a cluster-based thresholding technique, to control for family-wise error rate. We set an initial threshold of $p < 0.001$ and ran 100,000 permutations, while respecting the family structure of the partici-pants. $p$-Values related to each correlation matrix underwent FDR correction at q < 0.05. For all RSFC analyses, five participants were excluded from session 1 ($N = 937$ remaining) and four excluded from session 2 ($N = 865$ remaining) because of missing fluid intelligence scores.

### GSR

We were interested in the impact of GSR on our observations of time of day effects. To this end, we regressed the GS from the ICA-FIX denoised CIFTI data. We defined GS as the mean time series computed from all cortical grayordinates [37,38]. We then repeated our analyses of the effects of time of day on regional BOLD signal fluctuation and on RSFC.

### Control analyses

We compared the line of best fit for the effect of time of day on GS fluctuation using three types of regression analyses: OLS, robust regression, and quantile regression. The rationale was to observe the potential effect of heteroscedasticity, which should only influence OLS fit but not the other two approaches. We repeated these analyses for both between-participant and within-participant analyses. OLS and robust regression were computed using Scipy in Python, whereas robust regression was run in MATLAB (robustfit) with Huber weighting using the default tuning constant.

## Supporting information

**S1 Fig.** Scatterplots showing (A-C) effects of time of day on GS fluctuation, (D-E) effects of time of day on RV SD, and (G-I) effects of time of day on GS fluctuation after controlling for respiratory variation with colour-coding of data-point density. High density of data points around 12:30 PM is consistent with the planned timing of resting-state scans based on the HCP study protocol (HCP Reference Manual—1200 Subjects Release; Page 33). These same results are presented and described in more detail in Figs 1, 3 and S3, S4, S8 and S9 without colour-coding of data-point density. See S1 Data for underlying data. GS, global signal; HCP, Human Connectome Project; RV SD, standard deviation of respiratory variation. (TIF)

**S2 Fig. GS fluctuation as a function of time of day shown separately for each run across two sessions.** (A-D) Grey dots denote individual participants. Black dots show mean of GS fluctuation in hourly time windows. Line of best fit (red) was calculated based on data from all participants in each plot. Confidence interval is shown in light red. R values denote Pearson r correlation coefficient. *p*-Values were derived from 100,000 permutations while keeping family structure intact. GS fluctuation was defined as the standard deviation of the GS. See S2 Data for underlying data. GS, global signal; RV SD, standard deviation of respiratory variation. (TIF)

**S3 Fig. Negative association between time of day and GS fluctuation remains significant after controlling for RV SD in session 2.** (A) Between-participant variation of RV SD. (B) Run effects on RV SD at different times of day. (C) Between-participant variation of GS fluctuation residual as a function of time of day. (D) Run effects on GS fluctuation residual at different times of day. GS fluctuation residual was computed by group-level regression of RV SD from GS fluctuation. Grey dots denote individual participants. Black dots denote mean of GS fluctuation in hourly (left) or 3-hourly time windows (centre). Confidence interval is shown in light blue or light pink. R values denote Pearson r correlation coefficients. *p*-Values were derived from 100,000 permutations while keeping family structure intact. See S1 Data for underlying data. GS, global signal; RV SD, standard deviation of respiratory variation. (TIF)

**S4 Fig. Effects of time of day and of RV SD on GS fluctuation.** (A, B) Participants with greater RV SD (brighter dots) and scanned earlier in the day are more likely to exhibit greater

GS fluctuation than those with lower RV SD (darker dots) and those scanned later in the day (C) Participants scanned a longer duration apart on the two sessions (greater time of day Δ) are more likely to exhibit a greater between-session difference in GS fluctuation (greater GS fluctuation Δ) and in RV SD (greater RV SD Δ). Participants with greater RV SD on session 2 are denoted in red, whereas those with higher RV SD on session 1 are denoted in blue. (D-F) As expected, statistically controlling for the effects of RV SD on GS fluctuation via group-level regression eliminates the apparent visual gradient pattern along the y-axis, reflecting the systematic contribution of RV SD to GS fluctuation. These results are also presented in Fig 3 and S2 Fig without colour coding of RV SD. See S1 Data for underlying data. GS, global signal; RV SD, standard deviation of respiratory variation.
(TIF)

**S5 Fig. RSFC region is negatively correlated with time of day across participants in session 2 (*n* = 865), with a magnitude that surpasses the strength of correlation between fluid intelligence and RSFC.** (A) Correlation between time of day and RSFC across participants. (B) Correlation between fluid intelligence and RSFC across participants. Colours in lower triangular of correlation matrix denote z-transformed Pearson r correlation coefficients. Colours in the upper triangular denote z-transformed r values from the lower triangular averaged within network pairs. Colours on label axes denote correspondence of 419 regions to 17 large-scale cortical networks and to SC. Median absolute z values computed over the lower triangular were higher for time of day (0.13) than for fluid intelligence (0.04). Time of day–RSFC effects were significant, whereas RSFC–fluid intelligence effects were not significant for session 2, as assessed by network-based statistics (FDR-corrected at q < 0.05). Note that the colour scale for the fluid intelligence–RSFC effects was inverted to facilitate visual comparison with time of day effects. For session 1 results, see Fig 6 in the main text. FDR, false discovery rate; RSFC, resting state functional connectivity; SC, subcortex.
(TIF)

**S6 Fig. Global signal regression results in an overall reduction of negative correlations between time of day and regional BOLD signal fluctuation across the brain while introducing positive correlations at specific regions.** (A) Cortical regions showing significant correlations between time of day and BOLD signal fluctuation across participants in S1 (*n* = 942) and S2 (*n* = 869). (B) Subcortical regions showing significant correlations time of day and BOLD signal fluctuation across participants in S1 and S2. With the exception of brainstem, all subcortical regions are bilateral and presented as left-to-right hemisphere pairs (top to bottom). *p*-Values were derived from 100,000 permutations while keeping family structure intact. Colours (cool–warm) denote cortical and subcortical brain regions with significant z transformed Pearson r coefficients (q < 0.05, FDR-corrected), whereas nonsignificant regions are shown in grey. BOLD, blood oxygen level–dependent; FDR, false discovery rate; S1, session 1; S2, session 2.
(TIF)

**S7 Fig. Global signal regression reduces magnitude of negative correlations between RSFC and time of day while introducing positive correlations for several large-scale circuits in both session 1 (*N* = 937) and in session 2 (*N* = 865).** (A) Correlation between time of day and RSFC across participants. (B) Correlation between fluid intelligence and RSFC across participants. Levels of correlation are visibly stronger between time of day and RSFC than between fluid intelligence and RSFC. Colours in lower triangular of correlation matrix denote z-transformed Pearson r correlation coefficients. Colours in the upper triangular denote z values from the lower triangular averaged within network pairs. Colours on label axes denote

correspondence of 419 regions to 17 large-scale cortical networks and to SC. Time of day–RSFC effects were significant in both sessions as assessed by network-based statistics (FDR-corrected at q < 0.05), whereas fluid intelligence–RSFC effects were significant only in session 1. FDR, false discovery rate; RSFC, resting-state functional connectivity; SC, subcortex.
(TIF)

**S8 Fig.** Scatterplots showing (A-C) effects of time of day on GS fluctuation, (D-E) effects of time of day on respiratory variation, and (G-I) effects of time of day on GS fluctuation after controlling for respiratory variation. Error bars show standard error of hourly windowed means. These scatterplots are presented and described in more detail in Figs 1, 3, S3, S4 and S9, without the windowed standard error bars. See S1 Data for underlying data. GS, global signal.
(TIF)

**S9 Fig.** Scatterplots showing (A-C) effects of time of day on GS fluctuation, (D-E) effects of time of day on GS fluctuation on respiratory variation, and (G-I) effects of time of day on GS fluctuation on GS fluctuation after controlling for respiratory variation. Lines of best fit were computed using three different methods: OLS regression (red), robust regression (blue), and quantile regression (green). Robust regression and quantile regression were chosen because these two approaches are less susceptible to heteroscedasticity than OLS. The r and *p*-values shown are from the OLS regression. The same scatterplots are also presented in Figs 1, 3, S1, S3, S4 and S8. See S1 Data for underlying data. GS, global signal; OLS, ordinary least squares.
(TIF)

**S10 Fig. HRV (HR RMSSD) as a function of HR (HR mean).** Data are shown for participants with pulse oximetry traces with sufficient quality to enable reliable peak detection. For a participant to be included in session-level analyses, both runs had to pass quality criteria. Despite good-quality peak detection in these participants, there were some additional anomalies. In several runs, participants exhibited an HR of exactly 48 beats per minute, with very low levels of HRV, which were interpreted as likely artefactual in origin (shown in red). Other participants (shown in blue) were identified as likely having arrhythmia of nonsinus origin based anomalous distributions on Poincaré plots (not shown). Some of these participants are not visible on the plot because they have very high HRV (>200 RMSSD). The remaining runs were deemed suitable for analysis of cardiac data (shown in grey). See S1 Data for underlying data. HR, heart rate; HRV, heart rate variability; RMSSD, root-mean-square of the successive differences.
(TIF)

**S1 Data. Data underlying Figs 1 and 3 and S1, S3, S4, S8, S9 and S10 Figs.**
(XLSX)

**S2 Data. Data underlying Fig 2 and S2 Fig.**
(XLSX)

## Author Contributions

**Conceptualization:** Csaba Orban, Ru Kong, Michael W. L. Chee, B. T. Thomas Yeo.

**Data curation:** Csaba Orban, Ru Kong, Jingwei Li.

**Formal analysis:** Csaba Orban.

**Funding acquisition:** Michael W. L. Chee, B. T. Thomas Yeo.

**Methodology:** Csaba Orban, Ru Kong, B. T. Thomas Yeo.

**Resources:** Ru Kong, B. T. Thomas Yeo.

**Software:** Csaba Orban, Ru Kong, Jingwei Li, B. T. Thomas Yeo.

**Supervision:** B. T. Thomas Yeo.

**Validation:** Csaba Orban, Ru Kong, Jingwei Li.

**Visualization:** Csaba Orban.

**Writing – original draft:** Csaba Orban.

**Writing – review & editing:** Csaba Orban, Ru Kong, Jingwei Li, Michael W. L. Chee, B. T. Thomas Yeo.

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
