## [Editor Report · Decision Letter 0]

28 May 2019

Dear Dr Orban, 

Thank you for submitting your manuscript entitled "Time of day is associated with paradoxical reductions in global signal fluctuation and functional connectivity" for consideration as a Research Article by PLOS Biology.

Your manuscript has now been evaluated by the PLOS Biology editorial staff, as well as by an Academic Editor with relevant expertise, and I am writing to let you know that we would like to send your submission out for external peer review.

Please re-submit your manuscript within two working days, ie. by May 30 2019 11:59PM.

Kind regards,

Gabriel Gasque, Ph.D.,

Senior Editor

PLOS Biology

---

## [Decision Letter · Decision Letter 1]

2 Jul 2019

Dear Dr Orban,

Thank you very much for submitting your manuscript "Time of day is associated with paradoxical reductions in global signal fluctuation and functional connectivity" for consideration as a Research Article at PLOS Biology. Your manuscript has been evaluated by the PLOS Biology editors, by an Academic Editor with relevant expertise, and by three independent reviewers. You will note that reviewer 1, Matthew F. Glasser, has signed his comments.

In light of the reviews (below), we will not be able to accept the current version of the manuscript, but we would welcome resubmission of a much-revised version that takes into account the reviewers' comments. We cannot make any decision about publication until we have seen the revised manuscript and your response to the reviewers' comments. Your revised manuscript is also likely to be sent for further evaluation by the reviewers.

Your revisions should address the specific points made by each reviewer. Having discussed these comments with the Academic Editor, we think reviewer 3's point regarding task fMRI data is interesting, but probably out-of-scope. It feels more appropriate for a follow-up study. Thus, we do not expect this new analysis to be included in the revision. However, the robustness-related analyses mentioned later in the review are reasonable and should be taken seriously.

Please submit a file detailing your responses to the editorial requests and a point-by-point response to all of the reviewers' comments that indicates the changes you have made to the manuscript. In addition to a clean copy of the manuscript, please upload a 'track-changes' version of your manuscript that specifies the edits made. This should be uploaded as a "Related" file type. You should also cite any additional relevant literature that has been published since the original submission and mention any additional citations in your response. 

Before you revise your manuscript, please review the following PLOS policy and formatting requirements checklist PDF: http://journals.plos.org/plosbiology/s/file?id=9411/plos-biology-formatting-checklist.pdf. It is helpful if you format your revision according to our requirements - should your paper subsequently be accepted, this will save time at the acceptance stage.

Please note that as a condition of publication PLOS' data policy (http://journals.plos.org/plosbiology/s/data-availability) requires that you make available all data used to draw the conclusions arrived at in your manuscript. If you have not already done so, you must include any data used in your manuscript either in appropriate repositories, within the body of the manuscript, or as supporting information (N.B. this includes any numerical values that were used to generate graphs, histograms etc.). For an example see here: http://www.plosbiology.org/article/info%3Adoi%2F10.1371%2Fjournal.pbio.1001908#s5.

For manuscripts submitted on or after 1st July 2019, we require the original, uncropped and minimally adjusted images supporting all blot and gel results reported in an article's figures or Supporting Information files. We will require these files before a manuscript can be accepted so please prepare them now, if you have not already uploaded them. Please carefully read our guidelines for how to prepare and upload this data: https://journals.plos.org/plosbiology/s/figures#loc-blot-and-gel-reporting-requirements.

Upon resubmission, the editors will assess your revision and if the editors and Academic Editor feel that the revised manuscript remains appropriate for the journal, we will send the manuscript for re-review. We aim to consult the same Academic Editor and reviewers for revised manuscripts but may consult others if needed.

We expect to receive your revised manuscript within two months. Please email us (plosbiology@plos.org) to discuss this if you have any questions or concerns, or would like to request an extension. At this stage, your manuscript remains formally under active consideration at our journal; please notify us by email if you do not wish to submit a revision and instead wish to pursue publication elsewhere, so that we may end consideration of the manuscript at PLOS Biology.

When you are ready to submit a revised version of your manuscript, please go to https://www.editorialmanager.com/pbiology/ and log in as an Author. Click the link labelled 'Submissions Needing Revision' where you will find your submission record. 

Sincerely,

Gabriel Gasque, Ph.D., 

Senior Editor

PLOS Biology

Reviewer remarks:

Reviewer #1, Matthew F. Glasser: The authors present an interesting study of the effect of time of day scanned on global signal. I am not sure I understand the mechanism of the finding and it is not what I would have expected based on prior literature. Because the authors used HCP data and started from the recommended HCP preprocessing, their methods are generally of very high quality. I have a few minor comments. 

It will of course be important to replicate these findings after appropriate global noise removal, e.g. with temporal ICA, when that is available. I think it also would be interesting to extend the work to task fMRI data that has been appropriately cleaned for spatially specific and global noise to see if this pattern holds when subjects are performing a task. Additionally, it would be interesting to investigate which temporal ICA components are driving this phenomenon (I expect RC1 and/or RC5 from Glasser et al 2018 Neuroimage). If this is driven mainly by RC1, the phenomenon might not be seen in task fMRI data as RC1 is not present (and RC5 is much weaker). None of these data are yet publicly available for the authors to use; however. 

Minor comments:

1) It sounds like the authors carried out the laborious task of QC of the HCP’s physiological noise measures. Ideally this data would be shared back to the HCP for public distribution as recommended in Glasser et al 2019 Neuroimage. 

2) The use of a stringent FD threshold in HCP data is problematic for reasons discussed in Glasser et al 2018 Neuroimage and Power et al 2019 BioRXiv. A better motion measure would be dips and peaks after sICA+FIX cleanup in DVARS. 

3) The HCP structural acquisition was described in Glasser et al 2013 Neuroimage. 

4) The HCP structural preprocessing was a customized pipeline that included FreeSurfer along with other things.

5) I wouldn’t say “corrected for head-motion via a 24 parameter regression” as we know from Power’s work among others that movement regressors hardly correct for subject head motion and even doing the 24 parameter movement regression is becoming more controversial given its potential to remove neural signal while contributing modest additional denoising above sICA+FIX (Glasser et al 2019 Neuroimage). Better to just say that the movement regressors were regressed out.

Reviewer #2: This is study of the time of day on global signal fluctuations as well as regional fluctuations and connectivity is carried out on the Human Connectome Project dataset. The results are clear even though the authors fail to provide a compelling mechanism for this paradoxical effect. Changes in cortical excitability over the course of the day, at least to me, seems to be the most interesting explanation. The appropriate controls are performed although I believe that the nonlinear and spatially varying effects of respiration on GS may not be fully accounted for in their respiration effect normalization. In general, it is an interesting finding and a clear paper overall. 

Specific comments: 

In the abstract, the reference to a 1% to 3% BOLD signal change does not do much in putting a 22% decrease in context since the % BOLD change is of the raw signal and the 22% decrease is of a standard deviation measure. My guess is that the global standard deviation in terms of percent, shifted from 2% to 1.6%. 

Global signal fluctuations also include respiration effects, CSF pulsation, and brain pulsation. It may very well be that these could be reduced with dehydration as the day progresses. You mention appropriately, the possibility of fluid intake at the end of your paper.

It is observed that RV standard deviation was also correlated with time of day as well. It is mentioned that the correlation to time of day was a bit was weaker than that of GS std. Using linear regression to correct for this, the GS standard deviation effect remains. Linear regression to remove the effect may miss the nonlinear impact that respiration may have. It has been shown recently - in a talk at OHBM by Catie Chang - that the “respiration response function” first described by Birn et al actually varies considerably throughout the brain, so the respiration effect may still perhaps be having an impact on GS that is unable to be regressed out by simple linear modeling. 

On page 22 you mentioned: "These latter effects, however, might only become apparent once the global signal is removed from the data, for example, as was the case with the positive correlations between time of day and Somatomotor – Control network RSFC in the current study (Figure S5)” On the contrary, this might also be an artifact induced by GSR as Murphy et al argued. They showed that GSR induces artifactual correlations in resting state signals that were previously uncorrelated. 

Finally, in recommendations, I would suggest that you drill down a bit on how much these effects may influence other studies that likely have a distribution of time of day or a fixed time of day for their studies - evenly distributed across comparison populations. It’s not clear to me that it is something we really need to worry about unless we are studying one population in the morning and one in the evening. It may have a more significant impact on attempts to characterize individuals (i.e. fingerprinting). More discussion on these nuances would be appropriate.

Reviewer #3: In this study the authors look at the effect of time of day on the magnitude of resting-state fMRI global signal fluctuations and connectivity. They find that there is a decrease in both global signal magnitude and resting-state connectivity. While somewhat interesting, the effects are rather weak, accounting for less than 4% of the variance in the data. The title, abstract, and main text need to be significantly modified to acknowledge how weak this effect especially in comparison to the much stronger relations with other factors such as arousal that have been shown in prior studies. 

The study would also benefit from a consideration of other data that might shed light on the state of the subjects that are potentially related to arousal. For example, in the HCP dataset there are also task-based paradigms – an examination of the performance of the subjects on these tasks could provide some information on subject state. There should also be some consideration of how the fact that subjects were engaged in a lengthy fMRI experiment might have affected the diurnal variations. An fMRI experiment is a rather an unusual activity to partake in and it reasonable to expect that this could lead to a deviation from the “usual” variations. For example, anticipation of the exam might grow over the course of the day. This is an important point to address because the authors assume throughout their paper that the subjects are experiencing a “typical” decrease in arousal over the course of the day – this assumption gives rise to the “paradoxical” observation. However, this assumption is not adequately supported given the special circumstances involved in performing an experiment. 

The paper would benefit from a more through treatment of the considerable amount of scatter in the data. This may be partly addressed through better plotting of the data (see below). 

On a related note, there also needs to be a more detailed consideration of how large global signal fluctuation values may be driving the least squares fit and the windowed means. For example in Figure 1a,b there seem to be both fairly long tails of the GS values at each time of day and a great deal of temporal variability in the behavior of the tails. It would be good to verify that the scans with these large GS values exhibit reasonable behavior and to also examine the effects using robust estimators. Furthermore, an examination of the residuals is needed to determine the extent to which heteroscedastic effects are affecting the fit. 

Overall, a better analysis of outlier, leverage, and influence effects is needed. For example, in Figure 1C. most of the data appears to be centered about 0h, but the line is probably overly influenced by the relatively fewer observations at the extreme ends. Indeed if one to provide standard error estimates of the windowed values, then the standard error would be quite small around 0h and increase greatly as one goes to the extremes.

Additional comments

1. Abstract: it is misleading to say these results “challenge the prevailing notion that the brain’s global signal reflect mostly arousal and physiological artifacts” – while the authors show a rather weak effect, other studies show a very strong and robust relation between global signal and arousal and physiological artifacts. For example, the effects are particularly strong and repeatable as the subjects go from wakefulnesss to sleep. 

2. It is also misleading in the abstract to contrast the 22% decrease in the global signal with a 1 to 3% evoked BOLD responses, especially given the weak nature of the effect. There is enough sensitivity to detect the task-related response in a single scan and voxel, whereas the observed GS effect is only weakly seen even given a very large sample size. Also, the GS magnitude is on the order 0.2% so a 22% change corresponds to a change of only 0.04% in BOLD percent change units. This gives a more meaningful sense of the change. 

3. For the scatter plots, it would be useful to provide some indication of the density of the the points – for example, using something like scatplot or dscatter in MATLAB. 

4. Confidence intervals for the regression plots should also be provided. 

5. Standard errors should be displayed for the windowed estimates. 

6. It would be useful to demonstrate the combined dependence of GS magnitude on both time of day and RV SD. For example a 3D scatter plot of the data with time of day and RV as x and y axes and GS magnitude as z-axis may be interesting. 

7. It would be more meaningful to report the GS magnitude in terms of percent change BOLD signal. Presumably, the preprocessed data has used the HCP preprocessing that scales each 4D volume to a grand mean value of 10,000. If this is the case, then the global signal fluctuation magnitudes reported in Figure 1 are on the order of 0.2% which is consistent with prior work. 

8. P. 8 – it is stated that the correction for multiple comparisons (when considering different measures) is explained in the Methods, but this does not appear to be the case. The multiple comparisons explanation in Methods appears to be for the regional maps. 

9. Given the weak observed effects, the recommendations in section 4.9 seem a bit of an overreach – if the authors are to make such recommendations, it would be useful for them to provide an estimate of how much time of day would affect the conclusions of a typical fMRI study, which in general have much smaller sample sizes.

---

## [Decision Letter · Decision Letter 2]

2 Dec 2019

Dear Csaba,

Thank you for submitting your revised Research Article entitled "Time of day is associated with paradoxical reductions in global signal fluctuation and functional connectivity" for publication in PLOS Biology. I have now obtained advice from two of the three original reviewers (reviewers 1 --Matthew F. Glasser-- and 3) and have discussed their comments with the Academic Editor. 

Based on the reviews, we will probably accept this manuscript for publication, assuming that you will modify the manuscript to address the remaining points raised by the reviewers. Please also make sure to address the data and other policy-related requests noted at the end of this email.

We expect to receive your revised manuscript within two weeks. Your revisions should address the specific points made by each reviewer. Please submit a file detailing your responses to the editorial requests and a point-by-point response to all of the reviewers' comments that indicates the changes you have made to the manuscript. In addition to a clean copy of the manuscript, please upload a 'track-changes' version of your manuscript that specifies the edits made. This should be uploaded as a "Related" file type.

In addition to the remaining revisions and before we will be able to formally accept your manuscript and consider it "in press", we also need to ensure that your article conforms to our guidelines. A member of our team will be in touch shortly with a set of requests. As we can't proceed until these requirements are met, your swift response will help prevent delays to publication.

*Copyediting*

*Published Peer Review History*

*Early Version*

*Submitting Your Revision*

Sincerely,

Gabriel Gasque, Ph.D., 

Senior Editor

PLOS Biology

DATA POLICY:

In addition to your raw data from the Human Connectome Project, please provide as supporting information or upload to a publicly available repository the individual numerical values that underlie the summary data displayed in the following figure panels: Fig 1A-D, 2A-F, S1A-I, S2A-D, S3A-D, S4A-F, S8A-I, S9A-I, and S10A-D.

If you deposit your data in a publicly available repository, please provide the accession code or a reviewer link so that we may view your data before publication. 

Please ensure that the figure legends in your manuscript include information on where the underlying data can be found and ensure your supplemental data file/s has a legend.

Reviewer remarks:

Reviewer #1, Matthew F. Glasser: The authors have addressed my prior concerns. I would request a minor tweak of the future section to read “There are many avenues to extend the current study. For example, it will be interesting to explore whether the same effects can be seen during task-fMRI. In addition, Glasser and colleagues proposed the use of temporal ICA (Glasser et al., 2018) to decompose the fMRI data into multiple components, some of which appear to reflect “global” artefacts, which can then be more selectively removed, and some of which may relate to neural signals for arousal or eyes open versus closed. It would be interesting to investigate how these distinct global components might relate to time of day. Furthermore, some of these “global” components are present only during resting-fMRI, but not task-fMRI (Glasser et al., 2018). Thus, some of the effects we observe in this study might not appear in task-fMRI.”

Reviewer #3: The authors have made substantial revisions that largely address the prior concerns. I just have two remaining minor comments:

1) In the abstract, the authors now state: “These findings reveal unexpected effects of time of day on global brain activity that are not easily explained by arousal or physiological artefacts.” However, I think this is still misleading as the authors did not actually make any measurements of arousal state. I think what they mean to say is the “expected arousal” state. As noted in the prior comments and as acknowledged by the authors in the revised work, the actual arousal state may have differed from the expected arousal state due to a number of factors. 

2) For Figure S1, please add a colorbar to describe the color scheme used.

---

## [Editor Report · Decision Letter 3]

15 Jan 2020

Dear Dr. Orban,

On behalf of my colleagues and the Academic Editor, Dr. Ben Seyomur, I am pleased to inform you that we will be delighted to publish your Research Article in PLOS Biology. 

PRESS 

We frequently collaborate with press offices. If your institution or institutions have a press office, please notify them about your upcoming paper at this point, to enable them to help maximise its impact. If the press office is planning to promote your findings, we would be grateful if they could coordinate with biologypress@plos.org. 

Kind regards,

Krystal Farmer

Development Editor

PLOS Biology

on behalf of

Gabriel Gasque,

Senior Editor

PLOS Biology